# Characteristics and Injury Patterns of Road Traffic Injuries in Urban and Rural Uganda—A Retrospective Medical Record Review Study in Two Hospitals

**DOI:** 10.3390/ijerph18147663

**Published:** 2021-07-19

**Authors:** Selin Temizel, Robert Wunderlich, Mats Leifels

**Affiliations:** 1Department of Hygiene and Environmental Health, Augsburg University Hospital, 86156 Augsburg, Germany; 2Department of Anesthesiology and Intensive Care Medicine, University Clinic Tübingen, 72076 Tübingen, Germany; robert.wunderlich@med.uni-tuebingen.de; 3Singapore Centre for Environmental Life Sciences Engineering, Nanyang Technological University, Singapore 637551, Singapore; mats.leifels@rub.de

**Keywords:** road traffic crashes, road traffic injuries, Uganda, emergency medicine, LMIC, Sub-Saharan Africa, public health

## Abstract

In the ongoing Second Decade of Action for Road Safety, road traffic crashes pose a considerable threat especially in low-income countries. Uganda shows a vast burden of non-fatal injuries and resides at the top range of countries with the highest death rates due to unsafe roads. However, little is known about the differences in road traffic associated injuries between urban and rural areas and potential influence factors. Here, we used a cross-sectional study conducted by a retrospective medical record review from trauma cases admitted in 2016 to hospitals in rural and urban areas in Uganda. Injury severity scores were calculated and descriptive analysis was carried out while multivariate logistic regression was applied to assess significant covariates. According to the 1683 medical records reviewed, the mean age of trauma patients in the dataset under investigation was 30.8 years with 74% male. The trauma in-hospital mortality was 4% while prevalence of traumatic injuries is 56.4%. Motorcycle users (49.6%) and pedestrians (33.7%) were identified as the most vulnerable groups in both urban and rural setting while mild injuries of extremities (61.6%) and the head/neck-region (42.0%) were registered most. The frequency of road traffic injuries was homogenous in the urban and rural hospitals investigated in this study; interventions should therefore be intensified ubiquitously. The identification of significant differences in road traffic crash and injury characteristics provides the opportunity for specific programmes to decrease the socio-economic and health burden of unsafe roads. In addition to law enforcement and introduction of a Systems Thinking approach to road safety including infrastructural and educational concepts, the strengthening of trauma care and health resources is recommended.

## 1. Introduction

In 2010, the United Nations (UN) announced a Decade of Action for Road Safety to emphasize the burden of road traffic associated injuries (RTI) and fatalities globally [1]. The importance of RTI was further affirmed by their inclusion into the health targets of the Sustainable Development Goals (SDG) launched in 2015: Until 2020, a 50% reduction in road traffic deaths and injuries is to be achieved [1]. In August 2020, the United Nations Assembly rung in the second Decade of Action from 2021 to 2030 by reinstituting the goal to reduce RTI by 50% in the next 10 years [2]. In developing countries, injuries caused by road traffic crashes (RTC) claimed more lives annually than the leading transmittable and infectious diseases Malaria, HIV/AIDS and tuberculosis combined in the early 2000′s [3]. Of the 1.2 million global RTC fatalities, 90% occurred in low-income and middle-income countries (LMIC) even though less than half of all vehicles were registered there [4]. Without extensive efforts to tackle road traffic casualties, they are expected to rank at the fifth position of leading causes of death globally by 2030 with an ever increasing gap between poor and rich countries [5]. As early as 2012, RTI headed the mortality ranking among the age group of 15 to 29 in LMIC even though total vehicle associated deaths stagnated globally [6]. Measures to reduce road accidents have positive effects with four low-income countries (LIC) recording a decrease in road traffic fatalities in 2010 and Uganda was not among them [6]. In addition to deaths on the road, 50 million annual non-fatal RTI contribute to the global burden of disease with 23% of all global ‘Years lived with disability’ (YLD) [7]. The burden of non-fatal injuries and permanent disabilities due to trauma and RTC exceeds the number of deaths by approximately fifty times [3] even though non-fatal injuries are investigated far less than deaths [8].

Uganda, which is a sub-Saharan low-income country in East Africa with a population of 44 million in 2019, an annual Gross Domestic Product growth of 2,9% in 2020 and a life expectancy of 63 years in 2018 [9], registers up to 50 road traffic crashes each day with total of 18,426 affected according to statistics by the World Bank [9]. Of those affected by RTC, 17.5% die and 74.5% are injured so severely that they subsequently require inpatient treatment [10]. According to official accounts of the Uganda Bureau of Statistics, the number of people killed per 100 crashes recorded in the Accident Severity Index (ASI) amounted to 18 in 2015 with a total number of 3224 road deaths in that year [11]. By contrast, the World Health Organization (WHO) estimates a 3.5 times higher number of fatalities with a total of 10,280 deaths and 27.4 fatalities per 100,000 population since eligible death registration data are not available for Uganda [12]. Uganda therefore resides in the top range of countries with the highest death rates due to unsafe roads [6]. Even though more than two-third of those involved in a crash die before reaching a health facility [13], deaths by RTC caused 2.4% of hospital based mortality in 2014 and 2015, thus ranking fourth place in death causes among all ages and sixth place in deaths among age groups of 5 years and over [11]. The majority of the Ugandan population (84%) lives in rural areas [14] and while most traffic and subsequently most road associated crashes occur in the urban settings (52.5% of all RTC occur in the capital city Kampala), most fatalities (77.7%) are recorded outside of metropolitan areas [15].

The officially charted national Ugandan road network stretches over 20,544 km, of which 20% are paved with an additional 115,000 km of (mostly gravel and dirt) roads that spread on the district and community level [16]. Ninety percent of all Ugandan public and goods traffic takes place on those roads, resulting in considerable strain on unpaved road especially during the rainy season. While no official registry of motorised vehicles exists, between 700,000 and 1,200,000 vehicles are estimated on Uganda’s roads, with a double-digit growing trend especially in motorised two-wheel vehicles [16].

These figures further emphasise the importance of addressing RTI as a Public Health priority in LMIC in general and in Uganda in particular. However, for tailored prevention programmes, detailed knowledge on crashes and RTI epidemiology is crucial [17]. In similar geographical local and scientific contexts on the African continent, official statistics tend to be partially inaccurate and the data collection ends on the roadside while police statistics underestimated the burden of road deaths and injuries [18]. Various characteristics of RTI such as demographic differences in gender, age, vulnerable groups as well as road user and injury patterns have been addressed before to a certain extent in the past– even though recent data are limited, with a low double digit number of studies conducted in the past two decades [19].

Differences between urban and rural settings are rarely investigated in Uganda in terms of crashes and RTI even though availability of health-services and road conditions are well reported [20]. By filling this gap, this study aims to provide comprehensive epidemiological data, disclosure of shortcomings in RTI treatment and a robust foundation for more tailored road safety programmes to alleviate their socio-economic and health burden. 

## 2. Methods

### 2.1. Ethical Approval

Ethical approval was provided by the University of Liverpool Institutional Review Board (IRB) in Liverpool, UK, (reference UoL 201135969) and Hospitals in Kampala and Kasese, Uganda, involved in this study.

### 2.2. Study Design

A cross-sectional study has been conducted to investigate the prevalence and characteristics of injuries derived from road traffic (RTI) in two comparable and representative hospitals in rural and urban Uganda. The probability of prevalence calculation and the option to investigate more than one outcome variable at a time renders this design most feasible to answer the research question [21]. In order to extend the study group, a retrospective ‘nested’ data collection approach commonly known as a Medical Record Review (MRR) was chosen [22]. This well described and robust approach is a resource saving option (both monetary and timely) that allows retrospectively addressing a research question over a longer period. However, this tool has been associated with potential bias as the collection of valid data is dependent on the health care worker collecting and documenting the data during anamnesis of the patient involved in RTC as well as the researcher [22]. MRR has been shown to be a reliable tool for obtaining data for trauma investigation, especially when conducted in retrospective [23].

### 2.3. Setting

Secondary data has been collected from two hospitals catering to comparable patient numbers in rural and urban Uganda: Urban Hospital 1, located in Uganda’s capital of Kampala (population of 1.68 million in Kampala and 6.7 million in the metropolitan area in 2019 [24]) with more than 270 beds and 13,000 to 19,000 annual admissions between 2000–2015; Rural Hospital 2 in the provincial capital of Kasese in Uganda’s Western Province (population of 792,200 in Kasese Municipality in 2019 [24] and 75.5% of which are classified as rural according to the local municipal government [25]) with 200 beds and between 12,000 and 14,000 annual admissions in the same time frame. 

Both clinics are general hospitals receiving patients with all health issues in their catchment areas and represent—even though they are private non-profit hospitals—tertiary health facilities commonly found in Uganda. Due to organizational reasons, Rural Hospital 2 was chosen first and Urban Hospital 1 was selected as it was most comparable (and accessible) with respect to the main study parameters (i.e., number of trauma patients, number of beds and annual admissions) in the capital of Kampala. As maternal health is not related to trauma, admissions to the maternity and neonatal ward (8742 admissions in Urban Hospital 1 and 2284 in Rural Hospital 2 in 2016) have been excluded from the MRR and statistical calculations. The same is the case for outpatient care that were excluded due organisational reasons of the respective hospital record departments. The number of transferals from or to the hospitals under investigation was taken into consideration but it was investigated in detail by Werner, Lin [26] and Zheng, Sur [27] and lack of data due to incomplete patient files made it challenging to estimate and subsequently to ensure comparability.

### 2.4. Study Sample 

In order to ensure optimal coverage for the MRR, all eligible patients admitted to both hospitals between 1 January and 31 December 2016 have been included in this study (see Figure 1). For sample size calculation, the ‘Cohort and Cross-Sectional Studies’ feature of Epi Info™ 7.1.5.2 (Centres for Disease Control and Prevention; Atlanta, GA, USA) is utilized. Urban/rural are set as explanatory variables and RTI as outcome variable. The amounts of 49.0% and 64.2% of trauma related hospital admissions are caused by crashes on roads in urban and rural areas, respectively [17]. With this, a power of 80%, a confidence level of 95% and a sample size of 334 with an equal number of 167 participants in both groups are calculated. With a 20% buffer, 400 participants were determined to be necessary for the calculation of significance. 

The inclusion of a hospital administrative patient was determined by the following variables: gender, age, duration of stay, involvement in RTC and information on injury characteristics. Mode of transport and injured road user were not required for inclusion since they were inconsistently recorded (see Appendix A). The exclusion criteria were set as follows: trauma succeeds or precedes 1 January and 31 December 2016, incomplete data in inclusion criteria, admission due to secondary complications after trauma (e.g., osteomyelitis) and admission not being the first presentation for the experienced trauma in the study hospitals. 

Since late admissions due to trauma as well as pre-visits to traditional healers are common in Uganda [28], the first presentation in the hospital for permanent fixing was counted even though the trauma might lag for days or weeks. 

### 2.5. Instrument and Data Collection Methods 

During an on-site visit in March 2017 and April 2017, hospital admission data and patient’s records from the two hospital’s archives were reviewed in person for the collation of secondary data by the authors. All admission books of relevant wards and the study period were reviewed and eligible cases were recorded using Microsoft Office Access (Microsoft Corporation; Redmond, WA, USA) with a digital interface covering the study variables (see Appendix A). Cases in which the cause of admission was not determined (no information in the ‘diagnosis’ column) were recorded and subsequently investigated. Files of all recorded cases were reviewed on site for further data collection while general statistical data on the total admissions were obtained from the respective hospital administration. 

In accordance to best practice and European data privacy laws, a final identifier composed of a randomly assigned string consisting of digits and uppercase and lowercase letters was assigned to each patient in this study (Random.org, Trinity College, Dublin, Ireland). Duration of stay was noted in days whereby discharge on the same day as admission is counted as one (‘1’). Subsequently, a classification of short (1–3 days), moderate (4–9 days) and long (>10 days) was determined. Overall RTC outcome was distinguished between fatal and non-fatal since this study design does not allow predictions about, e.g., long-term disability. Crash specific variables were derived in accordance to WHO recommendations on injury severity and adapted in terms of mode of transport to fit the scope of the study. For data analysis, the differentiation between pedestrian, bicycles, motorcycle (if mentioned, involvement of a commercial ‘boda-boda’ motorcycle taxi), car including vans (the main mode of public transportation in Uganda), heavy transport vehicle and bus as mode of transport was recorded.

Injury location and type were adopted from the ICD-11 International classification of diseases coding in chapter XIX [12]. As presented in more detail in Appendix A, the categorization of injury types consisted of superficial wounds and deeper/open wounds, closed/open/not classified fractures, luxation/strain/sprain, blunt trauma/contusion and internal bleeding including brain oedema. Injury location is divided by location in the head, face, neck, spine, thorax, abdomen, upper arm including shoulder and scapula, lower arm including elbow, hand including wrist, upper leg including bony pelvis, lower leg including knee and foot including ankle. After on-site data collection, the injury severity was calculated using the Injury Severity Score (ISS), which a score adding up the Abbreviated Injury Scale (AIS) for multiple injuries with a minimum score of 1 and maximum of 75 [29]. The AIS defines severities (for the ISS 1 (mild) to 5 (critical)) for each injury condition. The three most severe AIS of the body regions head/neck, face, chest, abdomen, extremities and external were squared and summed up. For each body region, only the most severe injury was recorded.

### 2.6. Data Analysis

After completion of data collection, sensitive data were de-identified on-site and statistical analysis conducted using IBM SPSS, Version 21 (IBM, Armonk, NY, USA). Descriptive analysis of the variables in comparison between the urban and rural setting (group variables).

In the first part of the analytical section, differences between the two settings were investigated: For hypothesis testing (whether the differences between urban and rural were empirical), chi-squared tests were utilized for all categorical variables with more than five expected values (gender, RTC prevalence and mortality, mode of transport, road user, counterpart, injury location and type). For continuous and ordered categorical variables (age, duration of stay and ISS), Mann–Whitney U tests and Kruskal–Wallis tests were applied since all distributions are non-parametric (Shapiro–Wilk test of normality with *p* < 0.001). For all continuous variables, histograms were reviewed for kurtosis and skewness and mathematically determined; statistical significance is stated if value ÷ standard error > 1.96. In general, *p*-values < 0.05 were determined as significant. 

In the second part of the analysis, logistic regression was performed to investigate the influence of explanatory variables (age, gender, road user and urban/rural setting) on the probability of outcomes (injury outcome, duration of stay, ISS, injury location and injury type). Starting with bivariate logistic regression, direct dependencies were calculated (*p*-values < 0.05 determined as significant). Subsequently, combinations of confounding factors were tested via multivariate regression. 

## 3. Results and Discussion

### 3.1. Epidemiology of RTI

During the study period of 2016, 5018 eligible (i.e., non-maternity related) patients were admitted to Urban Hospital 1 and 9735 to Rural Hospital 2. Of these, 790 and 1247 were identified as trauma related resulting in a prevalence of 15.7% in Urban Hospital 1 and 12.8% in Rural Hospital 2, respectively. After application of the exclusion criteria, 688 trauma cases were eligible for data analysis in Urban Hospital 1 resulting in a dropout rate of 12.9%. For Rural Hospital 2, 995 complete data sets were available, thus resulting in a dropout of 20.2% (see Figure 1). Considering that similar studies faced crucial missing data such as cause of injury in most cases (up to more than 80% of the cases for Chokotho, Mulwafu [30]), data collection can be considered as highly successful. 

The share of road traffic associated injuries from all trauma cases accounted for an average of 56.5% (403 (58.6%) in the urban and 548 (55.1%) in the rural setting), meaning that the calculated minimal sample size of 200 crash related cases per hospital was exceeded two-fold. The difference of 3.5% was not significant (*p* = 0.15) and it could be demonstrated that unlike other studies, road traffic crashes contribute to an equal burden of disease in both rural and urban settings.

As shown in Figure 2, the mean age of trauma patients included in this study was 30.8 years (29.2 in the urban area and 31.7 years in the rural area, respectively), while the distribution showed a positive kurtosis (leptokurtic; k > 0) and a positive skewness (median < mean), which indicated that more young people are affected than the normal distribution would suggest. With a share of 30.4% of all cases, the age group from 25 to 34 years contained the most trauma patients, while more than two-thirds of all trauma patients were aged between 15 and 44 years, which are in line with previous studies in comparable socio-economical and geographical contexts [31]. 

With 74%, almost three-quarters of the study population were male (26.0% female) with no significant differences observable between the two hospitals neither for trauma patients in general nor for those seeking medical care for RTI (*p* > 0.05). Women were slightly more affected by crashes in the urban setting though (*p* = 0.07). 

### 3.2. Duration of Hospital Stay

The average duration of hospital stays was observed to be 6.3 days with a median of 4.0 days with a leptokurtic distribution, indicating that more people stayed for shorter periods expected from a normal distribution (see Appendix A). Trauma patients in the Rural Hospital 2 are admitted significantly longer (mean 7.1 days) than those in Urban Hospital 1 (mean 5.0 days) (*p* < 0.001). While the share of patients staying 4 to 9 days was identical for both settings (38.1%), patients at Rural Hospital 2 generally remained in-house longer than Urban Hospital 1 (18.2% of all patients reside for more than 10 days in Rural Hospital 2 compared to 10.6% in Urban Hospital 1). These differences remained significant in the group of patients involved in a crash (*p* < 0.001) but not among non-RTI (*p* = 0.33). 

### 3.3. Mortality

The overall mortality was calculated as 4% for all trauma cases (see Appendix A). This value was within the range of 2.7% and 6% stated by similar studies for general trauma mortality in Uganda [9] and 2.2 to 7.4% for road traffic injury associated mortality in other African countries [32]. For the period under investigation in this study, Urban Hospital 1 showed a significantly higher mortality rate than Rural Hospital 2 (*p* = 0.01). It has been observed before [13] that this anomaly in mortality can correlate with the distance of crash location to the hospital and the availability of ambulances, both of which has significantly reduced the time before an injured individual receives medical care [2,33].

### 3.4. Injury Severity 

The average person involved in a crash presented at both hospitals in the presented work was injured moderately with an ISS score of 9.9 (8 was the most frequent value). The differences in the ISS between both settings were evident: Even though most patients in urban/rural environments were categorised as mildly injured, Urban Hospital 1 showed a comparably increased number of injuries with higher severity. 

While more than two-third of the injuries in the rural setting could be classified as mild, the same share spreads among the three other categories observed for Urban Hospital 1 (see Figure 3). For severe and profound injuries, the difference was more notable: Urban Hospital 1 showed 15-fold more injuries categorised as more profound than Rural Hospital 2. Compared to other studies classifying 97% injuries classified as mild [34], this study displayed an increased rate in severe injuries. 

However, most comparable studies did not exclude outpatient consultations where a comparably higher share of mild injuries is to be expected. The differences in severity could also be influenced by the limited diagnostic abilities in the rural setting (i.e., absence of advanced imaging capabilities). This could explain the second peak in injury severity in the urban setting: Urban Hospital 1 has a computer tomography-scan and, subsequently, patients have been diagnosed with considerably higher number of severe head injuries.

### 3.5. RTC Specific Variables

For crash specific variables, the high rate of missing data in Rural Hospital 2 due to limited information in the medical records must be accentuated, especially for the accident counterpart; information was available in less than one-third of the cases (see Appendix A).

Overall, half the patients involved in crashes and admitted to the hospital were travelling on motorcycles, one-third were pedestrians and only 10% used a car. For vehicles, more passengers were presented to the hospital than were the drivers. In half of the recorded cases, the accident counterpart was logged as a car and then followed by motorcycles (41.9%) (see Appendix A).

As depicted in Figure 4, notable differences were observable within both urban and rural settings: Pedestrians were 27% more affected in the urban area, while motorcycles dominated rural crashes by 20% (*p* < 0.05). Passengers were 30% more affected in the rural setting (*p* = 0.003). For two-thirds of the RTC, the counterparts in the rural setting were riding motorcycles while the same share accounts for cars in Urban Hospital 1 (*p* < 0.001). Some differences could be explainable by infrastructural differences such as paving and sidewalks as well as traffic density, vehicles and (especially in rural areas) speeding [10]. Consistent with previous research findings, this study identified both motorcyclists and pedestrians as the most vulnerable groups. Compared to official statistics on accidents occurring on streets in Uganda, the results indicated a higher burden of these vulnerable groups [11].

### 3.6. Injury Characteristics

Almost two third of RTI-patients admitted to both hospitals showed injuries of the extremities (upper and lower), while more than half (55%) reported injuries of the head/face/neck/spine region and the thoracic/abdominal region was affected in a combined 14.2% of cases (19.1% in urban and 10.6% in the rural setting).

As visualized in Figure 5, several significant differences in injury location could be observed between the two settings: Head injuries were recorded twice as frequent at Urban Hospital 1 than in the Rural Hospital 2 (64.3% vs. 25.5%, *p* < 0.001). Spine injuries, thorax injuries and abdomen injuries were significantly more common in the urban setting, whereas a clear dominance was observed in the rural context for injuries in the lower extremities, especially of the lower leg (31.9% of patients at Rural Hospital 2 vs. 18.4% at Urban Hospital 1, *p* < 0.001). In terms of multiple injury locations, patients at Urban Hospital 1 showed elevated rates than compared to their rural counterparts (*p* < 0.001). 

Regarding the injury patterns, soft tissue injuries (STI) were noted in 63.3% of the crash patients with a slight increase in deeper wounds than superficial wounds. Half of all patients showed fractures of the extremities and nearly the same number sustained contusion of the head and other blunt trauma (see Figure 6). The differences observed for injury location were also reflected in the injury types: While fractures of the head and contusion were significantly more common in Urban Hospital 1 (73.4% of patients show contusion of which 21.8% had diagnosis of skull fractures compared to 25.4% and 1.8% in Rural Hospital 2, respectively, *p* < 0.001), fractures outside the head region dominated in the rural setting (54.7% vs. 46.2%, *p* = 0.01). Regarding polytrauma, more multiple injured patients were observed in the urban setting (*p* < 0.001). Most patients with RTI in Urban Hospital 1 showed two or more injuries, while only one injury is recorded with an almost linear decrease in Rural Hospital 2.

### 3.7. Influence of Covariates

Logistic regression was performed for statistically significant results from descriptive analysis to evaluate the potential influence of the investigated covariates including setting, gender, age and road user on the outcome variables that are fatality, duration of hospital stay, ISS, injury location and injury type (see Appendix A). Even though a statistically significant difference in total mortality between the urban and rural setting was observed (odds ratio (OR) = 1.828; meaning an 83% increased risk of dying because of a crash in urban than in rural Uganda), it was not observable in the group seeking medical attention for road traffic injuries. In logistic regression, none of the explanatory variables had a significant influence on the calculated differences (*p* > 0.05).

Urban patients with RTI stayed at the hospital significantly shorter (OR 1.372; 37.2% increased probability) while age and gender were determined to be additional influence factors on the duration of the stay. The group of patients > 65 years (for which the proportion was higher in Rural Hospital 2) stayed in the hospital longer than younger age groups, while males stayed longer in the hospital than females. RTI were found to be significantly more serious in the urban setting: The odds ratios for being seriously or profoundly injured and admitted to Urban Hospital 1 rose to 5.25 and 23.06, respectively (see Appendix A). Gender influenced the injury severity as well: Females displayed milder injuries more frequently while males dominated the group of the moderately injured (*p* < 0.05).

For the four most important significant differences in road traffic injury locations (head injuries and thoracic injuries as well as upper and lower leg), the influence of several covariates was identifiable: Comparing head injuries, age and mode of transport showed an influence on the injury location even though the difference between the settings remained significant with a 5-fold increased risk for head injuries at Urban Hospital 1 (OR = 5.242). Children aged 0 to 14 years, adults from 45 to 54 years and pedestrians showed a higher rate of head injuries after RTC than the other age groups. For thoracic injuries, the difference between the urban and rural setting was not tenable after adjusting for covariates; mode of transport (more injuries in car, heavy transport and bus users) especially proved to have a higher influence on thoracic injuries than the setting. For injuries of the lower extremities, no influence of the measured covariates was observable: RTI patients at Rural Hospital 2 have a more than 2-fold increased risk of suffering from an injury of the lower extremity when compared to those in Urban Hospital 1 (OR = 2.441).

### 3.8. Limitations of the Study

To our knowledge, the presented work was the first study with a focus on assessing the characteristics of injuries associated with road traffic crashes in a pair of representative urban and rural hospitals in Uganda. However, several limitations need to be addressed: First and foremost, with only two hospitals assessed, the generalisability of the results is limited. In this context, the non-profit but private nature of the assessed hospitals also requires consideration in terms of a potential (economic) selection bias [35]. Second, due to the low quality of data and records in certain categories such as Crash Counterpart (see Appendix A) and when compared to similar studies [36], patients observed at the out-patient department were not included in the data collection and thus potentially excludes a high number of trauma patients that might have an influence on the calculations of, e.g., prevalence, and resulting in a higher proportion of severe injuries. Mulago Hospital, the main referral hospital of the country logs admittance due to trauma at only 38% [17].

The pre-hospital emergency care in Uganda has been reported to be rudimentary at best [34] and the high number of trauma deaths outside the hospital might contort the findings. Furthermore, late presentation at the hospital (e.g., after having consulted a traditional healer) can result in a skewed picture of injuries sustained during a crash [13,33]. The same could be the case for the overall absence of private clinics and general practitioners in the rural setting. Their unavailability results in hospitals being the only choice for health services in general, which in turn results in a broader variety of (minor to severe) cases presented here [37].

Other important limitations originate from the retrospective MRR nature of the study that is inherently inflexible and does not allow for the analysis of variables and covariates such as occupation, social status, education or helmet use which might influence the conclusion of the findings but were not recorded during hospital admittance. Additionally, suspected diagnosis cannot necessarily be verified by gold-standard methods in LMIC and thus potentially results in information bias.

### 3.9. Implications of the Findings

On average, 56.5% of trauma associated hospital admissions in the presented work were related with road traffic crashes (see Appendix A). Previous lower rates in rural areas have not been confirmed in the presented work [8]. Age and gender of patients suffering from RTI in the study population correspond with previous studies as the most vulnerable group in terms of total numbers and the group was identified to be young males on commercial and privately-owned motorcycles [38]. The retrospective nature of MRR allowed for detailed insights into previously underreported sub-populations of those with RTI: in the age group 0 to 14, more than three-quarters of road traffic injury affected were pedestrians and also most road users > 55 years travelling on foot (see Appendix A). For these age groups, future measures to reduce RTI should consider focusing on this mode of transport—especially in urban areas where structure road-crossings via traffic lights or overpasses are oftentimes absent.

In regards of gender, males dominated those affected by injuries obtained during crashes in all modes of transport: men account for 94% of bicycle injuries and 75% of motorcycle injuries and showed equal risk for RTI regardless of an active or passive traffic role (pedestrian, driver or passenger). Females are almost at the same risk of being injured as a pedestrian and as a motorcycle-passenger, which is a vehicle they almost never actively drive [10].

### 3.10. Recommendations

Ugandan code of law includes a high rate of road safety rules and regulations, including speed limits, laws defining driving under the influence of substances (DUI) with low accepted levels of blood alcohol and drugs, helmet and seat belt obligations and mobile phone use [13]. Law enforcement levels ‘on the ground’, however, are generally estimated as low [6]. Preventive measures, nonetheless, depend strongly on intensive and reliable implementation of road safety laws which—in the case of Uganda—had previously been shown to be the most cost-effective measure [39]. Without appropriate enforcement and penalisation, other preventive actions such as public education have been shown to be less effective in similar contexts [40].

In terms of prevention (‘pre-crash’), focusing on vulnerable groups with infrastructural changes (e.g., safe pavements, street crossings and appropriate lighting) for pedestrians (especially vulnerable groups such as children and the elderly) and (motor-) and cyclists (e.g., by separated motorcycle lanes) was recommended by others [41]. Increasing the visibility of road users, e.g., by distributing reflectors to children and daytime light for vehicles, is recommended by the WHO and the UN and has been successfully implemented in the majority of 37 studies investigated by Kwan and Mapstone [42].

Continuous public traffic education, particularly for school children and commercial motorcyclists, has also demonstrated to raise the awareness for better safety behaviour among younglings and their caregivers. Initiatives by local shareholder, such as the ‘SafeBoda’ project which aims at educating and equipping commercial motorcyclists that represent a large share of private-hire transportation in Uganda, presented a particularly promising approach [10]. In the same context, precise observation of injury characteristics indicated other starting points for prevention in the crash-phase: Comprehensive helmet use for drivers and passengers is long known to reduce head injuries and strict enforcement of helmet laws already in place could decrease the number and severity of most common RTI in both rural and urban Uganda [41].

An example for locally driven approaches was the Uganda Helmet Vaccine Initiative that investigated socio-cultural barriers of helmet use and successfully implemented interventions to tackle these impediments and reduce prejudices against helmet use [43]. As women have been shown to be highly vulnerable as motorcycle passengers in this study and others, the question of their sitting position on motorcycles (both legs on one side with the risk of garment getting entangled in the wheel-spokes) and alternatives could be proposed for further investigation.

Due to the long history of research in the public health aspects of road traffic, most recommendations on the improvement of individual safety derived from post-crash measures. In the past two decades, various studies revealed that no nationwide emergency hotline exists in Uganda, that less than 11% of those seriously injured in a road traffic crash receive professional pre-hospital care and that specialised training for emergency medicine is not or not ubiquitously available, thus resulting in insufficient first aid administered on-site [6].

In addition to public health interventions targeting the affected group or individual, it is also recommended that improved surveillance of trauma trends in rural and urban hospitals by generating standard operating protocols for patient files and their medical history should be facilitated [44]. While generally suitable and well within the recommended parameters, the tool of MRR ultimately relies on a scoring system that improves with the quality of data available for individual patients [45]. Generalised anamnesis would greatly improve the ability of researchers, the hospitals and the Uganda Department of Health to conduct national surveys, which could then form the foundation for the most optimal allocation of funds into education [46].

In addition to the obvious necessity of addressing these shortcomings, first-aid courses for lay persons may improve the outcome of injuries caused by crashes as shown in pilot studies in Uganda and throughout Africa.

## 4. Conclusions

As early as 1998, road traffic crashes were referred to as ‘a worsening global disaster destroying lives and livelihoods, hampering development and leaving millions in greater vulnerability’, which is particularly the case in LMIC and Sub-Sahara Africa. Direct costs derived from RTC in LIC have been calculated to affect between 1 and 3% of the gross national product, either by medical treatment costs or (indirectly but no less severely) by loss of income during hospital stays or due to long-term injuries [47].

The expenditure associated with crashes in LMIC has been shown to exceed the amount of the total developmental aid in countries categorized as such by the World Bank [38]. The same study revealed that the socio-economical dimension of injuries associated with road traffic was vastly extensive as most of those involved in crashes were male and were young and thus associated with the societal group often responsible for the revenue of extended families. O’Hara and Mugarura [48] calculated that adults with long-bone fractures in Uganda lose up to 88% of their annual income, which potentially decreases the economic outlook of relatives. For Ghana, Abagale and Akazili [49] determined that the cost for direct and indirect medical treatment of injuries associated with crashes in the family exceeds the mean annual income of the lowest income group families by far. This results in significant burdens which poor families without health insurance can hardly overcome in a country without proper social safety net and it does not even account for the individual social tragedies and long-term effects of disabilities. Additionally, treating RTI occupies vast health system resources (desperately needed for other scourges such as malaria or mother and child mortality in most LIC and LMIC) and thus further impedes development [40].

Therefore, a scenario in which no avoidable road deaths occur (i.e., Vision Zero) should be prioritized in national and provincial policy and road safety and injury prevention programmes should be spawned. Prior to this ambitious goal, however, comprehensive and robust logistical, medical and societal interventions, including the improvement of RTI management and the involvement of policy makers, strengthening of local institutions and NGOs, should be aimed at.

## 5. Highlights

Trauma patients are mostly male (74%), 30.8 years old and show injuries of extremities and neck/head;Motorcycle users and pedestrians are the most vulnerable in both urban and rural setting;Frequency of injuries associated with road traffic crashes was homogenous in urban and rural hospitals;Context specific programmes recommended to decrease burden of unsafe roads.

## Figures and Tables

**Figure 1 ijerph-18-07663-f001:**
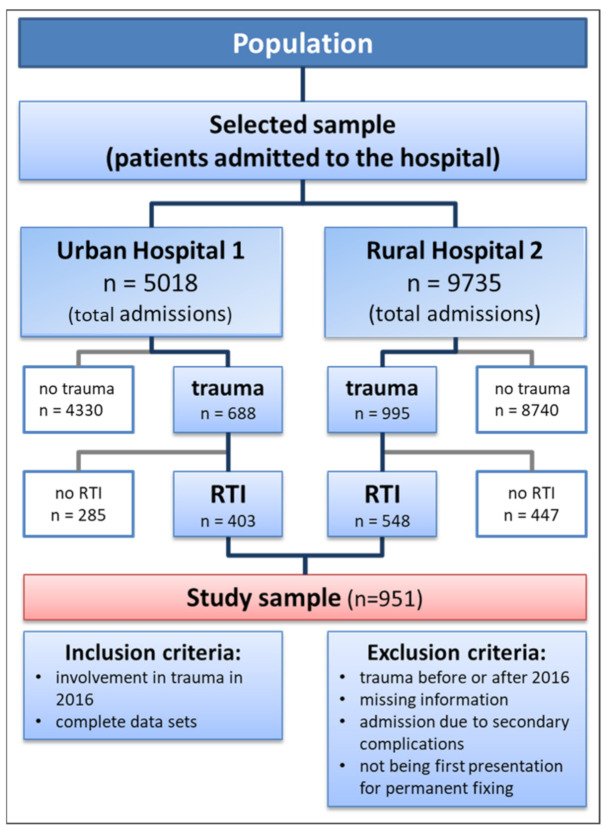
Population included in the study after excluding admissions that were not eligible for the definition of trauma and RTI in both hospitals.

**Figure 2 ijerph-18-07663-f002:**
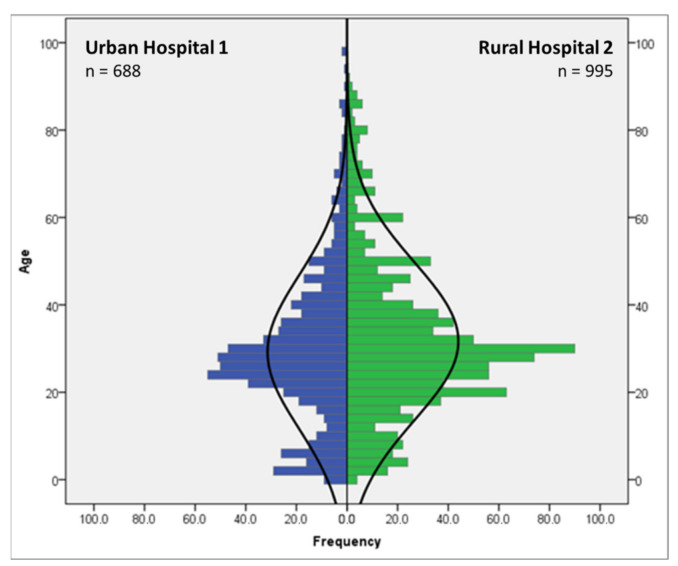
Age distribution of eligible trauma patients in the study population treated in Urban Hospital 1 and Rural Hospital 2.

**Figure 3 ijerph-18-07663-f003:**
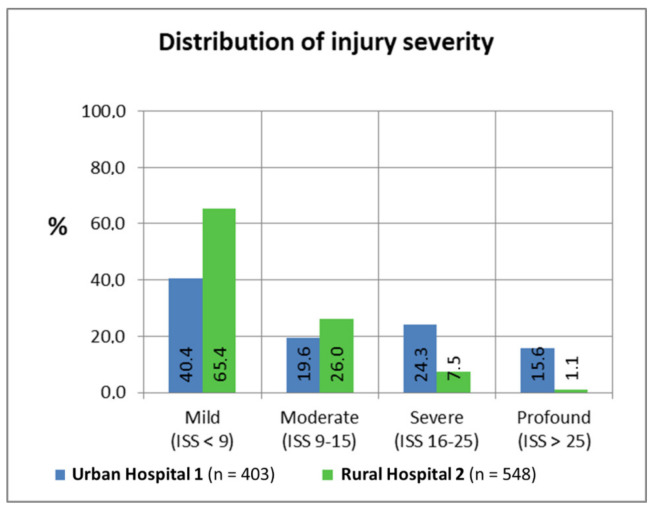
Distribution of Injury Severity Score (ISS) in the study population treated in Urban Hospital 1 and Rural Hospital 2.

**Figure 4 ijerph-18-07663-f004:**
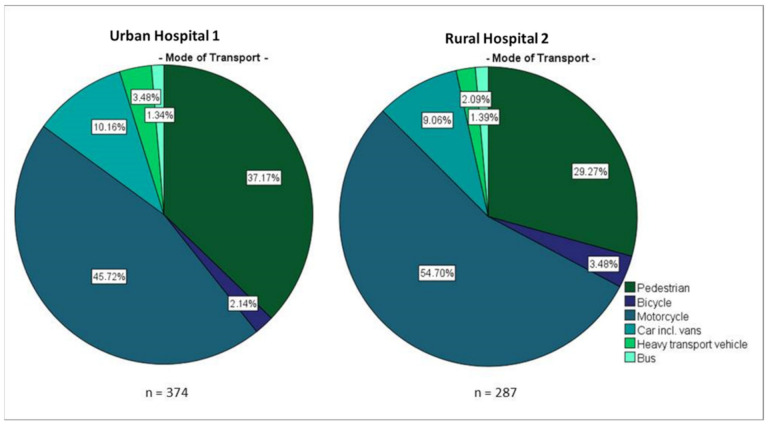
Mode of transport of the study population treated in Urban Hospital 1 and Rural Hospital 2.

**Figure 5 ijerph-18-07663-f005:**
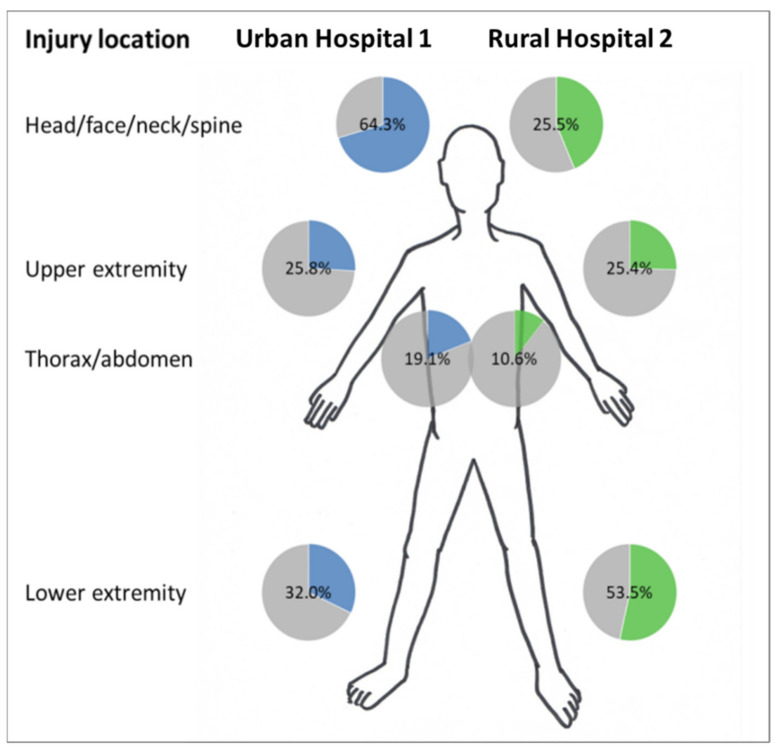
Distribution of injury location in the study population treated in Urban Hospital 1 and Rural Hospital 2.

**Figure 6 ijerph-18-07663-f006:**
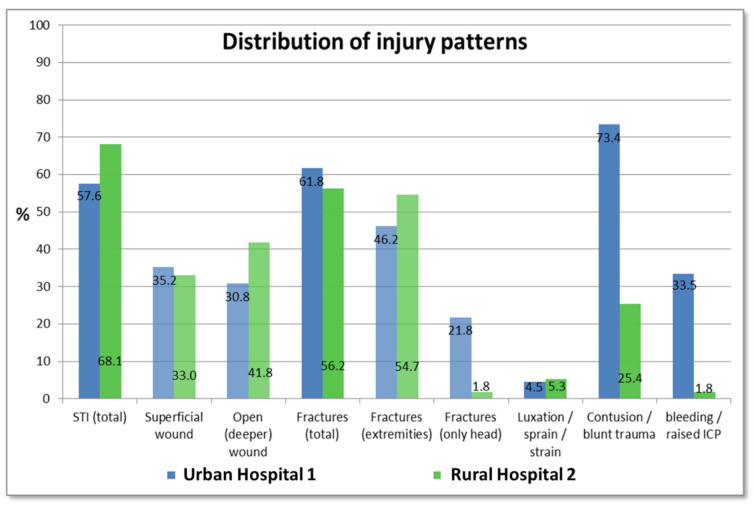
Distribution of injury patterns in the study population treated in Urban Hospital 1 and Rural Hospital 2.

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
