# Peer review of "Characteristics and Injury Patterns of Road Traffic Injuries in Urban and Rural Uganda—A Retrospective Medical Record Review Study in Two Hospitals"

_ijerph, 2021, doi:10.3390/ijerph18147663_

Round 1
Reviewer 1 Report
- Methods
(lines 117-122)
In this study, only the hospitals in which the study was conducted are described. For the readers, it needs further explanation on how these hospitals were selected for the study based on the general situation of hospitals or health care facilities in Uganda.
- Results and Discussion
(line 214-222)
Readers would understand the sample of the study easier if the paragraph(lines 214-222) is presented with Figure 1.
(lines 228-235)
Is the population distribution of urban and rural areas the same in Uganda?
Was the difference in the population distribution by region considered for the analysis?
(lines 254-259)
How can it be interpreted that there is no difference in mortality in urban and rural areas while there is a difference in severity?
(lines 275-281)
Have you considered whether it is the first hospitalization or transfered from other hospitals? Is there any possibility that the difference between urban and rural areas is due to the hospital transfers?
(Implication)
What are the recommendations after reflecting the differences in duration and injury severity in urban and rural areas?
Can the frequency of road traffic injuries in urban and rural areas be homogeneous given a general difference in the number of the population and the traffic?
Author Response
Review Report Form
- Methods
(lines 117-122)
In this study, only the hospitals in which the study was conducted are described. For the readers, it needs further explanation on how these hospitals were selected for the study based on the general situation of hospitals or health care facilities in Uganda.
Thank you for raising this very important point that has not been addressed sufficiently in the original draft. The choice of hospitals has been in part due to access and rural Hospital 2 was selected first, and urban Hospital 2 chosen subsequently as it was most comparably in an urban setting. More information has been added in lines 125 – 128.
“Due to organizational reasons, Rural Hospital 2 was chosen first and Urban Hospital selected as it was most comparable (and accessible) in regards of the main study parameters (i.e. trauma patients, number of beds and annual admission) in the capital of Kampala.”
- Results and Discussion
(line 214-222)
Readers would understand the sample of the study easier if the paragraph (lines 214-222) is presented with Figure 1.
Thank you for this suggestion, Figure 1 was moved to the first paragraph of the Results and Discussion to improve accessibility of the study size for the reader.
(lines 228-235)
Is the population distribution of urban and rural areas the same in Uganda?
Was the difference in the population distribution by region considered for the analysis?
Thank you very much for raising this point, we spent quite some time discussing it before and during the on-site work. While the population dynamics in Uganda have changed vastly since its independence in 1962 and the first official assessment by the Ugandan bureau of statistic and the World Bank (95.53% of a population of 6,76 million was classified as rural in then compared to 75,64% of 41,64 million in 2019), the largest portion of Ugandans is still living outside of areas that are defined as urban (World Bank, 2021). According to the latest projections for 2020 by the United Nations and the Uganda Bureau of Statistics, populations of the four Regions are relatively evenly distributed (UBos, 2019):
Central (including Kampala): 11,562,900 (40,580 km² Area, 284.9/km² Population Density in 2020)
Eastern: 10,836,500 (31,810 km² Area, 340.7/km² Population Density in 2020)
Northern: 8,606,300 (85,290 km² Area, 100.9/km² Population Density in 2020)
Western: 10,577,900 (51,340 km² Area, 206.0/km² Population Density in 2020)
Therefore, and considering the main intention of the study, we concluded that even though the cultural and socio-economic conditions in the four regions may differ and the impact of the Covid19 pandemic is yet to be evaluated, a hospital in the Kasese district could be considered representative for rural Uganda and compared to one in the capital of Kampala.
Additionally, we added additional information in line 119-120 and 122-123 (highlighted in blue):
Urban Hospital 1, located in Uganda’s capital of Kampala (population of 1,68 million in Kampala and 6,7 million in the metropolitan area in 2019 (UBos, 2019)) with more than 270 beds and 13,000 to 19,000 annual admissions between 2000 – 2015 as well as Rural Hospital 2 in the provincial capital of Kasese in Uganda’s Western Province (population of 792,200 in Kasese Municipality in 2019 (UBos, 2019), 75.5% of which are classified as rural according to the local municipal government (Kasese Municipal District, 2021)) with 200 beds and between 12,000 to 14,000 annual admissions in the same time frame.
(lines 254-259)
How can it be interpreted that there is no difference in mortality in urban and rural areas while there is a difference in severity?
Thank you for this observation and for your remark. It has been reported before that the absence of ambulance services and organized transportation of injured individual result in a significantly higher number of road traffic accidents with “fatalities” (i.e. the person suffering an injury dies on site due to lack of adequate medical care), which most likely resulted in the observation reported here (Alanazy et al., 2019).
Furthermore, some internal injuries such as trauma affecting organs and the brain could not be diagnosed in the rural hospital due to the inability to conduct sophisticated imaging. Subsequently, it is very likely that the severity of an injury was underreported and thus appeared as less severe in the results. This assumption has been discussed in the limitation section (Line 387 – 392).
Additionally, we added line 270 – 273 to elaborate on your remark:
“It has been observed before (13) that this anomaly in mortality can correlate with the distance of accident location to the hospital and the availability of ambulances, both of which has significantly reduces the time before an injured individual receives medical care (2, 33).“
(lines 275-281)
Have you considered whether it is the first hospitalization or transferred from other hospitals? Is there any possibility that the difference between urban and rural areas is due to the hospital transfers?
You are completely right, the risk of patients not seeking medical treatment directly after an accident, limiting their visit to the outpatient department or general physicians / traditional healers or being transferred from other hospitals leads to a potentially high amount of cases which are overlooked and not included into the presented work. We discussed these limitations in lines 387 – 397 and tried to present and discuss socio-cultural practices such as visiting traditional healers before hospital and overall conditions found in Uganda, that have been mentioned by others in the same context (Alanazy et al., 2019; Bayiga Zziwa et al., 2019; Hsia et al., 2010; Kobusingye et al., 2002).
(Implication)
What are the recommendations after reflecting the differences in duration and injury severity in urban and rural areas?
Can the frequency of road traffic injuries in urban and rural areas be homogeneous given a general difference in the number of the population and the traffic?
Thank you for raising this most important point in drawing any conclusions from the patient files under investigation in the manuscript. The main implication and recommendation from the data obtained in the rural and urban context is, that more qualitative research is needed to formulate individual and most suitable public health interventions and educational approaches. Programs such as helmet and seatbelt initiatives could be beneficial for both settings, but their acceptance and acceptability depend strongly on choosing sensitive communication and outreach to plant the message into the groups most affects. Without a robust scientific foundation, the risk of investing into unsustainable schemes has been demonstrated numerous times (Berg, 2006).
It is therefore encouraging to see that successful initiative such as safe-boda or the erection of overpasses resulted in slow but steady progress in the reduction of the burden of disease associated with transportation (Mutto et al., 2002; Vaca et al., 2020) and therefore the injury severity and hospital stay duration in both urban and rural clinics.
We tried to raise these points in 3.10. Recommendations in lines 422 – 465 and based on some studies and public health measurements that were reported and implemented during or after the sampling period the patient files presented here.
Reviewer 2 Report
The paper addresses an important public health topic. However, I think the paper's overall quality in terms of its writing clarity and style, organization of the information presented, and the flow of logic is unsatisfactory to keep it in the review process for further consideration. I recommend that the authors seek an experienced researcher's extensive advice regarding the preparation of a manuscript to be submitted to a peer-reviewed scientific journal in this field.
Author Response
Thank you for your time and for agreeing to review the submitted manuscript. We have revised draft according to comments by two other reviewers and sincerely hope that the changes and additions might address some of the limitation your identified in the presented work.
Reviewer 3 Report
This is a very well written and detailed paper. Not too many concerns but additional limitations should include:
Lower severity in the rural hospital could be the result of
A: Because of poor transport severe cases day prior to admission
B: The hospital may be the only choice for health services thus they see a broader range of severe to minor cases, whereas the urban hospital may act a referral centre for higher acuity cases
A recommendation for standardized incident reporting including something on the circumstances of trauma should be included.
Author Response
Comments and Suggestions for Authors
This is a very well written and detailed paper.
We sincerely thank you for your kind words and for your valuable comments.
Not too many concerns but additional limitations should include:
Lower severity in the rural hospital could be the result of
A: Because of poor transport severe cases day prior to admission
We completely agree with this suggestion, the correlation between the increase of “on-site road traffic fatalities” and the subsequent reduced number of mortalities in rural hospitals has been added in lines 270 – 273.
“It has been observed before (13) that this anomaly in mortality can correlate with the distance of accident location to the hospital and the availability of ambulances, both of which has significantly reduces the time before an injured individual receives medical care (2, 33).”
B: The hospital may be the only choice for health services thus they see a broader range of severe to minor cases, whereas the urban hospital may act a referral centre for higher acuity cases
We again agree and added the following sentences in line 397 – 401:
“The same could be the case for the overall absence of private clinics and general practitioners in the rural setting. Their unavailability results in hospitals being the only choice for health services in general which in turn results in a broader variety of (mi-nor to severe) cases presented here (37).”
A recommendation for standardized incident reporting including something on the circumstances of trauma should be included.
Thank you for raising this point. The limitations of the Medical Record Review as a tool for comparison when patient records differ in-between hospitals nationally or internationally has been discussed for some time now. For MRR to assess trauma in particular, this becomes an issue when the anamnesis does not go into the level of detail required to calculate scores according to systems like the Injury Severity Score chose here. It would therefore help future research, the hospitals and potentially the healthcare system of a country to initiate standardized formats that are either proposed by the governmental side (top-down) or by the hospitals in a “grassroot” kind of fashion.
We elaborated on this aspect in lines 467 – 475:
“Besides public health interventions targeting the affected group or individual, it would also be recommended to facilitate an improved surveillance of trauma trends in rural and urban hospitals by generating standard operating protocols for patient files and their medical history (44). While generally suitable and well within the recommended parameters, the tool of MRR ultimately relies on a scoring system that im-proves with the quality of data available for individual patients (45). Generalized anamnesis would greatly improve the ability of researchers, the hospitals, and the Uganda Department of Health to conduct national surveys which could then form the foundation for the most optimal allocation of funds into education-(46).”